# The Potential of Plant Polysaccharides and Chemotherapeutic Drug Combinations in the Suppression of Breast Cancer

**DOI:** 10.3390/ijms252212202

**Published:** 2024-11-13

**Authors:** Omowumi O. Adewale, Patrycja Wińska, Adrianna Piasek, Joanna Cieśla

**Affiliations:** Chair of Drug and Cosmetics Biotechnology, Faculty of Chemistry, Warsaw University of Technology, 00-664 Warsaw, Poland; omowumi.adewale@uniosun.edu.ng (O.O.A.); patrycja.winska@pw.edu.pl (P.W.); adrianna.zalewska.dokt@pw.edu.pl (A.P.)

**Keywords:** breast cancer, chemotherapy, plant polysaccharides, synergy, mechanism

## Abstract

Breast cancer is the most common cancer affecting women worldwide. The associated morbidity and mortality have been on the increase while available therapies for its treatment have not been totally effective. The most common treatment, chemotherapy, sometimes has dangerous side effects because of non-specific targeting, in addition to poor therapeutic indices, and high dose requirements. Consequently, agents with anticancer effects are being sought that can reduce the side effects induced by chemotherapy while increasing its cytotoxicity to cancer cells. This is possible using natural compounds that are safe and biologically active. There are many reports on plant polysaccharides due to their bioactive and anticancer properties. The use of plant polysaccharide together with a conventional cytotoxic drug may offer wide benefits in cancer therapy, producing synergistic effects, thereby reducing drug dose and, so, its associated side effects. In this review, we highlight an overview of the use of plant polysaccharides and chemotherapeutic drugs in breast cancer preclinical studies, including their mechanisms of anticancer activities. The findings emphasize the potential of plant polysaccharides to improve chemotherapeutic outcomes in breast cancer, paving the way for more effective and safer treatment strategies.

## 1. Introduction

Breast cancer, lung cancer, and colorectal cancer account for half of the newly diagnosed cancers in women, and among them, breast cancer represents the most frequently diagnosed category of cancer among women worldwide. According to the World Health Organization, breast cancer was the most common cancer in women in 157 countries out of 185 and caused 670,000 deaths in 2022 [1]. It is the most common cause of cancer-related death in most countries, accounting for one in four female cancer diagnoses and one in six cancer-related fatalities. It was also reported to be ranked first in terms of incidence [2]. Breast cancer incidence is higher in developed countries (55.9 per 100,000) than in developing countries (29.7 per 100,000), with the highest rates in Australia, New Zealand, Western Europe, North America, and Central America [1,3,4]. It is predicted that one in eight American women will be diagnosed with breast cancer during their lifetime. An estimated 310,720 women will be diagnosed with invasive breast cancer in 2024. However, these values are rising in developing countries [5,6,7], causing an increase in the global burden of breast cancer.

Age is one of the major risk factors to breast cancer incidence globally, with the highest rates seen in the oldest females. For instance, in the United Kingdom, over a third of breast cancers occurs in females above 70 years, while less than one in five women are under 50 at diagnosis [8]. In contrast, in less developed countries, over half of breast cancers occurs in females under 50. This is due to the younger population and shorter life expectancy, which most likely will change as life expectancy increases alongside economic development [1,5]. 

Apart from age, other important risk factors that influence breast cancer incidence are grouped into reproductive and non-reproductive factors, which are influenced by economic development. The risk of breast cancer rises with lower age at menarche, higher age at menopause, number of children, and duration of breastfeeding [9,10]. 

Risk factors belonging to the non-reproductive categories include obesity, increased alcohol consumption, and genetic mutations. The risk of breast cancer is observed to be doubled in overweight post-menopausal females, while increased alcohol consumption is estimated to account for approximately 4% of all diagnosed breast cancer cases in 2020 [11,12]. Genetic or hereditary factors such as *BRCA1* or *BRCA2* mutations have been shown to contribute to approximately 5–10% of breast cancers [13]. The lifestyle factor, among others, has also been listed among risk factors of breast cancer [7].

This study, aiming to determine the 25-year trend of the breast cancer mortality rate in seven super regions defined by the Health Metrics and Evaluation, has shown that in general, there is a significant increase in the breast cancer mortality rate in the world, which could be due to the increase in incidence and prevalence of this cancer. This is true especially for developing countries and low-income regions that experience sharp slopes in their breast cancer mortality rates [14].

During the past several years, our understanding of the complexity and heterogeneity of breast cancer has been significantly improved by advances in diagnostic methods including molecular profiling and imaging techniques. This has allowed for more accurate diagnoses and targeted treatment approaches. Despite these, there are still significant challenges with respect to resistance to treatment, toxicity, and patient outcomes. The constant limitations associated with conventional therapies, for instance chemotherapy, radiation, and surgery, highlight the pressing need for new therapeutic approaches to improve treatment efficiency while weakening side effects. Considering this, plant-derived polysaccharides are considered promising adjuncts to conventional therapies, availing synergistic effects to enhance treatment outcomes and also reduce chemotherapy-induced toxicities.

Although different approaches for treatment are constantly emerging, drug resistance and side effects are challenges that remain critical hindrances [15]. Breast cancer subtypes, complex genetic interactions, lifestyle, and environmental factors constantly confuse choices of treatment and prognostic outcomes [16]. All these additionally highlight the importance of a search for new approaches, particularly those that combine conventional therapies with natural agents like polysaccharides, which may synergistically boost efficacy while mitigating the associated toxicities with current therapies. 

Taking this into account, the development of synergistic strategies in breast cancer therapy is extremely critical in the reformation of cancer treatment models [17]. Complementary modes of action for various therapeutic agents can be harnessed by such strategies which should allow for an increase in anticancer efficacy while simultaneously reducing or removing associated adverse effects [18]. Furthermore, such approaches may help in overcoming drug resistance which is an intimidating obstacle in breast cancer management [19]. Combination therapies, especially chemotherapy with other conventional treatments, have become an especially important aspect of breast cancer treatment. Although each of the treatment modalities, such as surgery and chemotherapy, plays a significant role in cancer treatment, their combination is intended to achieve better tumour control and improve patient outcomes and prognosis. Surgical treatment is usually used to remove localized tumours along with the surrounding tissue, but there is usually a need to remove residual tumour cells with chemotherapy [20]. Equally, radiotherapy has been used as an important adjuvant or neoadjuvant in the treatment of tumour to prevent risk of local recurrence [21]. However, despite the advantages of these therapeutic combinations, they are still not without limitations. While each individual part of these combinations often induces systemic toxicity, the cumulative toxicities of the combination therapy may also aggravate treatment-related side effects, leading to treatment interruptions and dose reductions, and consequently, compromising therapeutic efficacy [22]. Hence, while a combination of chemotherapy with conventional treatment is a hope provided in breast cancer treatment, efforts to mitigate the associated side effects continue to be paramount in optimizing patients’ outcomes and their quality of life. In light of this, focus has shifted to natural products.

Natural agents, especially those derived from plant sources offer essential bioactive qualities, which can assist the mechanisms of action of chemotherapeutic drugs [23]. These natural compounds including polyphenols, flavonoids, and polysaccharides exhibit various beneficial effects, like antioxidative, immunomodulatory, and anti-inflammatory activities [24], that enhance the sensitivity of cancer cells to chemotherapy as well as modulate chemotherapy-induced toxicities [25]. Among these natural agents, plant polysaccharides have in the past decade stolen the basic cancer researchers’ attention because of their inherent medicinal qualities. Plant polysaccharides possess unique molecular structures and wide-ranging bioactivities that make them exceptionally useful for various medicinal purposes. Polysaccharides like β-glucans and pectins are known for their immunomodulatory, antioxidant, and anti-inflammatory properties which may help in enhancing the sensitivity of cancer cells to chemotherapy while protecting normal tissues from damage. Their natural origin also avails them an advantage that make them safe with little or no harmful side effects compared to conventional adjunct therapies, hence making them promising candidates in abating the challenges like toxicity and drug resistance in breast cancer treatment. They possess a long history of therapeutic potential in the treatment and management of diseases [26,27]. Through an exploration of the therapeutic potential of plant polysaccharides in combination with chemotherapy, the limitations associated with conventional treatments may be averted, consequently optimizing treatment potency as well as enhancing the patients’ quality of life [28]. Furthermore, associated cytotoxic effects of chemotherapy can be upgraded through the synergistic interactions with polysaccharides, hence enhancing the overall effectiveness of breast cancer therapy [29]. 

## 2. Chemotherapy Used in Breast Cancer Treatment

Chemotherapy involves the use of cytotoxic drugs in the elimination of rapidly dividing cells including cancer cells [30]. Chemotherapeutic drugs used in treatment of breast cancer consist of several categories of cytotoxic agents, such as alkylating agents, antimetabolites, anthracyclines, and tubulin inhibitors [31,32,33].

### 2.1. Alkylating Agents

Alkylating agents are among the primary and most used anticancer drugs. Most alkylating agents share similar mechanisms of action but vary in their therapeutic effectiveness. Alkylating agents with anticancer properties have efficacy throughout all stages of the cell cycle and are employed in the treatment of diverse range of malignancies [20]. Frequently, these drugs exhibit greater efficacy in the treatment of slow-growing malignancies, such as solid tumours (e.g. breast cancer) and leukaemia [34,35]. Alkylating agents function via three distinct mechanisms resulting in the disruption of DNA function and eventual cell death: i.By the generation of cross-links between atoms within the DNA structure—during this action, two bases are connected by bifunctional alkylating agents. Crosslinking hinders the separation of DNA strands, thereby preventing the DNA synthesis or transcription. On the other hand, if the bases are on the same DNA strand, it leads to strong attachment between the drug and the DNA molecule. This will not inhibit DNA strand separation but will inhibit the interaction of replication or transcription enzymes with the DNA, thereby blocking cell growth or leading to the induction of apoptosis.ii.By induction of mutations in DNA strands by causing nucleotide mispairing.iii.By induction of the alkyl groups’ attachment to DNA bases—subsequently, attempts by DNA repair enzymes to replace such alkylated bases lead to DNA fragmentation.

Examples of alkylating agents are cyclophosphamide, ifosfamide, and chlorambucil (Figure 1a–c), which are nitrogen mustard derivatives that induce crosslinking and fragmentation of DNA strands [36]. Cisplatin and carboplatin (Figure 1d and e are platinum-based alkylating agents [37].

### 2.2. Antimetabolites

Antimetabolites are cytotoxic agents that have been under development for over 50 years. They are structural analogues of either purine or pyrimidine bases or the corresponding nucleosides, or structural analogues of folate cofactors, which play a role in various stages of purine and pyrimidine biosynthesis [38]. The antimetabolite action results in inhibition of the synthesis of DNA constituents. On the one hand, antimetabolites induce depletion of nucleotide pools through the inhibition of enzymes involved in nucleotide synthetic pathways, and on the other, as structural analogues of purine and pyrimidine bases, they are incorporated into DNA or RNA instead of correct nucleotides, which leads to abnormal nucleic acids. Additionally, folate analogues can inhibit the enzymes using folate co-enzymes in numerous reactions [39,40]. Examples of common antimetabolites used in breast cancer therapy include 6-mercaptopurine, fludarabine, 5-fluorouracil, gemcitabine, cytarabine, and methotrexate (Figure 2). 

### 2.3. Anthracyclines

Anthracyclines are pharmaceutical compounds derived from *Streptomyces* species that are employed in the therapeutic management of many forms of malignancies. The possible therapeutic options are daunorubicin, doxorubicin, epirubicin, idarubicin, mitoxantrone, and valrubicin (Figure 3) [41]. Anthracyclines are highly potent chemotherapeutic drugs that effectively treat a wide range of malignancies, including breast cancer [42]. The mechanism of actions of anthracyclines involves different events; these include reactive oxygen species (ROS) production, membrane interruption, lipid peroxidation, and enzyme interactions. 

#### 2.3.1. ROS Generation 

Anthracyclines, through redox reactions, are capable of producing ROS. Anthracyclines have special chemical structures that can be converted by reduction to a semiquinone form through the catalytic action of nicotinamide adenine dinucleotide phosphate oxidase and nitric oxide synthases in the cell cytoplasm [43,44] and by electron transport chain in the mitochondria. This can result in the transport of electrons to the anthracycline such as daunorubicin to form semiquinone daunorubicin. The latter is an unstable metabolite and therefore could undergo oxidation through mitochondria-released oxygen, followed by the generation of ROS. The surplus ROS eventually leads to oxidative stress, membrane-lipid peroxidation, and DNA damage, thereby inducing apoptosis in the cell [45,46]. 

#### 2.3.2. DNA Intercalation 

Anthracyclines have a chromophore component that serves as an intercalating agent. This moiety inserts itself between neighbouring base pairs of DNA within the nucleus of a cell. This insertion of the chromophore component of anthracyclines into the groove of DNA helical structure causes some molecular consequences, including (i) disruption of the DNA double helical structure, thereby, affecting its stability and molecular function, and (ii) DNA and RNA synthesis inhibition, where the intercalation of the DNA strand by anthracycline prevents the DNA and RNA polymerase from properly binding to the DNA templates. As a result, it hinders the synthesis of both DNA and RNA, particularly in cells that undergo rapid replication. Consequently, it obstructs the process of cell division [46]. 

#### 2.3.3. Topoisomerase Inhibition

Anthracyclines also disrupt the activity of topoisomerase II, resulting in the formation of breaks in the DNA strands. Topoisomerase II is the enzyme responsible for the relaxation of the DNA torsional strain during replication, when the activity of this enzyme is disrupted, it is trapped at the point of DNA breaks, leading to a ternary complex consisting of the DNA with double strand breaks, topoisomerase II, and the anthracycline chemotherapeutic drug. Therefore, though the torsion strains in the DNA is relaxed, the resultant breaks cannot be rejoined. The creation of the ternary complex hinders the reconnection of the double-stranded DNA breaks. Consequently, it induces cellular growth arrest and triggers programmed cell death [47,48,49,50]. 

### 2.4. Tubulin/Microtubule Inhibitors

Microtubule inhibitors (MTIs) are a kind of medication that impede the normal functioning of cellular microtubules. Microtubules, composed of tubulin monomers, are vital components of the cellular cytoskeleton. Microtubules take part in several biological activities such as transport, cell structure, migration, and mitosis. MTIs work by either destabilizing or stabilizing the microtubules. Microtubule destabilizers interact with the central portion of the tubulin subunit, thereby preventing the polymerization of these subunits into microtubules. Microtubule stabilizers interact with tubulin and stabilize the microtubules, thereby preventing its disassembly and leading to cell cycle arrest. Examples of MTIs are taxanes, such as docetaxel and paclitaxel (Figure 4) that work by attaching to microtubules, stopping them from breaking down, which causes the cell cycle to halt and leads to programmed cell death [51].

## 3. Chemotherapy Challenges 

Prolonged usage of alkylating agents and other forms of chemotherapy have been associated with several detrimental side effects, such as inducing lifelong infertility by reducing sperm production in males and causing the cessation of menstruation in females. Many chemotherapeutic drugs have the potential to cause secondary malignancies, including acute myeloid leukaemia, several years after the completion of the therapy [52,53,54,55]. Chemotherapy is also linked to hair loss, weight alterations, nerve problems, sexual dysfunction, and anaemia [56]. Additionally, significantly high doses of chemotherapy are required to elicit its treatment, thereby increasing patients’ exposure to the adverse effects. Also, because these drugs are not specific for the target site, healthy tissues are also exposed, hence leading to dangerous unwanted side effects [56,57]. Though many different breast cancer therapies individually have potencies in the treatment of the disease to an extent, synergistic approaches have led to improved efficacies [58,59,60,61] and such an approach is usually the practice. Combination therapy in breast cancer involves the use of two or more therapeutic modes to successfully cure breast cancer. The combined effects of the numerous breast cancer therapies are explored in order to increase effectiveness, decline the likelihood of recurrence, and curtail resistance in comparison to a single therapeutic approach. Some common combination therapy in breast cancer include (i) chemotherapy in combination with another chemotherapy (polychemotherapy), (ii) chemotherapy in combination with hormonal therapy, and (iii) chemotherapy in combination with immunotherapy [62,63,64,65]. Consequently, there is a strong effort to improve the sensitivity or potency of chemotherapeutic agents, while reducing their harmful side effects. In this trend, scientists look for possible novel combinations with other types of anticancer drugs; however, due to the overall toxicity of chemotherapy, there is a growing interest in plant polysaccharides as substances that can aid breast cancer treatment.

## 4. Benefits of Plant Polysaccharides 

Plant polysaccharides are polymeric compounds composed of several identical or distinct monosaccharides components linked with alpha (α-) or beta (β-) glycosidic bonds [66]. They include starch, cellulose, pectin, and other similar compounds. Due to their widespread occurrence, the molecular composition and molecular weight of plant polysaccharides vary among various species. Polysaccharides derived from plants have gathered considerable interest in recent decades due to their notable bioactivities and suitability for therapeutic purposes [67,68]. The research has also proven that plant polysaccharides are non-toxic and devoid of any adverse effects [69,70]. 

The pursuit of novel chemical entities of natural origin as potential anticancer agents with no toxic side effects has resulted in the investigation of plant polysaccharides from a variety of sources. In vitro and in vivo studies over the past decade have demonstrated that numerous plant polysaccharides have promising anticancer activity against a variety of cancer cell lines and in animal models. Many of these agents have been reported to successfully address the main drawbacks of conventional chemotherapies [71,72,73]. Plant polysaccharides can selectively induce cytotoxicity in tumour cells without the commonly observed adverse effects [74,75]. The application of plant polysaccharides in the prevention and treatment of cancer has been connected to their wide range of striking biological activities.

### 4.1. Biological Activities of Plant Polysaccharides

Plant polysaccharides are endowed with striking bioactivities, and these have allowed them to be used as adjuncts to classical chemotherapy in cancer treatment [22]. It has been shown that the structure or components of polysaccharides are closely linked to their functional activities [76,77].

For instance, Wang et al. reported the structural analysis of hetero-polysaccharide, APSID3, extracted from *Astragalus membranaceus* to consist of one terminal arabinose, one 1,5-linked arabinose, one 1,3-linked rhamnose, one 1,3,4-linked rhamnose, five 1,4-linked methyl galacturonates, and six 1,4-linked methyl glucuronates [78]. The biological activities, e.g., immunomodulatory effects, of this polysaccharide have been reported both in vivo and in vitro. It has been documented that this polysaccharide enhances the functionality of dendritic cells, macrophages, T cells, B cells, and lymphocytes [79].

Among the primary bioactivities of plant polysaccharides, the following most especially, make them relevant for their implications in cancer therapy. 

#### 4.1.1. Antioxidative Activity

Antioxidants are compounds that inhibit oxidation, which is a chemical reaction capable of producing free radicals [80]. Free radicals are very reactive and unstable chemical species that can operate as oxidants or reductants by abstracting an electron from another molecule or by donating electron to them [81]. Free radicals usually have beneficial roles in the cells, for instance in cell signalling, immune response, detoxification, and adaptation to stress, which is controlled by cellular antioxidant system [82]. However, excessive production of free radicals overwhelms the antioxidant system and results into oxidative stress, which is implicated in several diseases [83]. The most important oxygen-containing free radicals in many diseases, including cancer, comprise the hydroxyl radical, superoxide anion radical, hydrogen peroxide (H_2_O_2_), oxygen singlet, hypochlorite, nitric oxide (NO) radical, and peroxynitrite radical. These are highly reactive species that can cause damage to the vital cell components such as DNA, proteins, carbohydrates, and lipids, causing cell damage and disrupting homeostasis [84,85]. The role of endogenous and exogenous antioxidants therefore becomes imperative in many oxidatively related diseases. Various plant polysaccharides have been reported to increase the cell’s capacity to scavenge free radicals, thereby reducing oxidative stress responsible for pathological changes [86]. Additionally, natural polysaccharides also exert antioxidant effects by regulating the signal transduction pathway and/or the activity of enzymes. For instance, some important antioxidant signalling pathways [87,88] were affected in rat chondrocyte and dendritic cells upon treatment with *Angelica sinensis* and *Cyclocarya paliurus* polysaccharides, respectively. The nuclear factor-E2-related factor 2 (Nrf2) signalling pathway is one of the major pathways being regulated by natural polysaccharides. Under normal conditions, Nrf2 is usually bound to Kelch-like ECH-associated protein-1 (Keap1) in the cytoplasm and Keap1 acts as Nrf2 portion sensor, which constantly promotes the rapid degradation of Nrf2, thereby maintaining its low levels in the cytoplasm. However, the introduction of polysaccharide antioxidants leads to the release of Nrf2 from Keap1, permitting its translocation to the nucleus. In the nucleus, Nrf2 causes the transcription of antioxidant response element (ARE), leading to the expression of downstream antioxidant proteins and phase II detoxification enzymes, which protect cells from oxidative stress [89]. Examples of polysaccharides known for regulation of Keap1/Nrf2/ARE signalling include *Lycium barbarum* polysaccharide and sulfated *Cyclocarya paliurus* polysaccharides (CPP) [88,90,91]. Other signalling pathways, such as the PI3K/AKT and JNK/IRS1 pathways, have also been reported to be influenced by some polysaccharides to further reduce oxidative damage. These include *Coptis chinensis* and wheat germ polysaccharides [88,92,93].

Furthermore, some polysaccharides have also been shown to exert significant antioxidant effects via modulation of the activities of key enzymes that are involved in the cellular antioxidant defence mechanism against oxidative stress [93]. For example, superoxide dismutase (SOD), catalase (CAT), and glutathione peroxidase (GSH-Px) comprise the first line of defence against ROS [94]. SOD acts by converting deleterous superoxide anions (products of electron transport chain in the mitochondria) into less harmful H_2_O_2_; this is eventually broken down into non-toxic water molecules by CAT and GSH-Px, hence preserving the cells from intended oxidative damage. The role of plant polysaccharides on these enzymes is based on their influence on the Keap1/Nrf2/ARE pathway leading to the expression of the genes of these antioxidant enzymes [95,96]. Moreover, plant polysaccharides can hinder the activities of oxidases including inducible nitric oxide synthase (iNOS) that promotes excessive production of ROS, an important contributor to oxidative stress, hence mitigating oxidative stress. Zhuang, et al. reported how pretreatment with *Angelica sinensis* polysaccharides caused a significant inhibition of iNOS activity in H_2_O_2_-exposed human osteoarthritis chondrocytes, consequently mitigating oxidative stress-induced injury [87]. The strong antioxidant properties of natural polysaccharides are due to this dual action, which increases the activity of antioxidant enzymes while blocking pro-oxidant enzymes. In the context of free radical scavenging capacity, plant polysaccharides such as those obtained from *Silphium perfoliatum*, *Ocimum sanctum*, *Aloe vera, Avicennia Marina*, *Moringa oleifera*, *Ginkgo biloba*, and so on were shown to possess significant free radical reducing power (e.g., cupric and ferric ions) and considerable nitric oxide, ABTS, and DPPH radical scavenging activities [97,98,99,100,101,102], thereby inhibiting oxidative stress.

The antioxidative capacity of plant polysaccharides has largely been associated with extraction methods, composition of monosaccharides, and molecular size. These are important factors identified to play a major role in influencing their mechanism of antioxidative activities [101,103]. It was reported that extraction condition involving at least 15 mL of water per gram of plant material at 97 °C for at least 60 min and lyophilization enhanced the antioxidant activities of some plant polysaccharides [102,103]. Also, tea polysaccharides with molecular weights between 2.31 kDa and 10.88 kDa were revealed to possess significant antioxidant activities [104], Meanwhile, Kang et al. revealed that *Aloe vera* polysaccharides rich in rhamnose and arabinose exhibited strong antioxidant action against intracellular ROS production and cell death [98]. Wu et al. also proved that the significant antioxidative activities observed for polysaccharide from *Silphium perfoliatum L.* is correlated with the content of uronic acid [101].

#### 4.1.2. Immunomodulatory Activity

Another important bioactivity of plant polysaccharides is their immunomodulatory activity. The ability of plant polysaccharides to modulate the immune system response has been widely reported [105,106]. These compounds can modify the immune response, hence enhancing its ability to fight against foreign pathogens and various diseases including cancer [107]. Polysaccharides with immunomodulatory effects are otherwise known as biological response modifiers [108]. 

Numerous studies have demonstrated that plant polysaccharides modulate the action of the immune system through various mechanisms and at different stages. In addition to the activation of immune cells such as T and B lymphocytes, macrophages, and natural killer cells, they also activate complement and stimulate the generation of cytokines exerting a regulatory effect on the immune systems [108,109,110]. The control of the innate immune system plays a crucial role in the capacity of the host to promptly respond to infections. Macrophages, being vital components of the host immunological defence system, can cooperate with other cell types, such as neutrophils, to combat external pathogenic substances [108,109,110,111]. The primary mechanism by which plant polysaccharides exhibit their immunomodulatory effects is firstly through their influence on macrophages. Plant polysaccharides induce macrophage stimulation (macrophage activation) for the production of ROS, the release of cytokines, and their phagocytic function. They can also promote the proliferation of immune cells [108]. For instance, among polysaccharide fractions extracted from *Styela plicata,* one fraction, unabsorbed on the DEAE-Sepharose CL-6B column (SF-1 fraction), caused a dose-dependent rise in the production of NO in RAW 264.7 cells [112]. NO is a significant bioactive compound that plays a crucial role in several physiological and pathological processes in biological systems, e.g., transmission of signals, immunological responses, and inflammatory reactions [113,114,115]. Within the immune system, the activation of macrophages leads to the generation of a substantial quantity of nitric oxide. NO has the ability to eliminate bacteria, parasites, and tumour cells, while also triggering inflammatory responses and safeguarding the body against external threats. Similarly, polysaccharide extracted from *Basella rubra* L., *Ixeris polycephala*, [116] and *Pterospartum tridentatum* (L.) Willk. [117] reportedly induce NO production in vitro in macrophages [108].

Plant polysaccharides also exert their immunomodulatory activity by influencing the macrophages to release cytokines. Cytokines are small proteins that play important roles in the regulation of cell–cell interactions, cell growth, and cell differentiation. Cytokines can be classified based on function as interleukins (ILs), tumour necrosis factors (TNFs), interferons (IFNs), and colony-stimulating factors (CSFs). They play important roles in the regulation of inflammatory responses and immune responses, and they affect both innate and adaptive immunity [108,118,119]. 

The sulphated polysaccharide fraction from *Styela plicata*, SF-1, aside from its influence on macrophage-induced NO production, also caused a dose-dependent release of cytokines, including tumour necrosis factor alpha and interleukin 6 but preserved the viability of RAW 264.7 cells [112]. 

Other plant polysaccharides that have successfully influenced the release of cytokines from immune cells include the polysaccharide component *from Citrus unshiu* that significantly modulates the secretion of TNF-α and IL-6 (the pro-inflammatory cytokines), and IL-12 (the anti-inflammatory cytokine) in macrophage RAW264.7 [120]. 

This suggests that plant polysaccharides have the ability to modulate the production of both pro- and anti-inflammatory cytokines. For example, IL-12 works as a negative feedback mechanism to avoid excessive activation of macrophages in the hyper-inflammatory response. Plant polysaccharide components from *Alchornea cordifolia* reduce the secretion of cytokines, including IL-1β, IL-6, IL-10, TNF-α, and GM-CSF, in human and mouse macrophages cultured in vitro [121].

#### 4.1.3. Anti-Inflammatory Activity

Inflammation is an initial reaction of the body’s immune system that occurs when the body experiences injury, infection, or stress [118]. Typically, inflammation is a normal defence mechanism accompanied by the release of nitric oxide and pro-inflammatory cytokines. However, if inflammation persists for an extended period of time, it may be harmful and contribute to the development of diseases including cancer [122,123,124,125]. Hence, natural products with anti-inflammatory ability show beneficial effects in the inhibition of prolong inflammation, hence showcasing their anticancer mechanism. They utilize specific modes of action to produce their anti-inflammatory effects:(1)They can suppress the release of substances that cause inflammation or regulate the imbalance between anti-inflammatory cytokines (IL-10), and pro-inflammatory cytokines (IL-1β, IL-6, and TNF-α) [126].(2)Plant polysaccharides can also exert their anti-inflammatory effect by regulating the NF-κB-associated signalling pathway.

The nuclear factor kappa B (NF-κB) pathway is a crucial pathway that plays a significant role in inflammation. NF-κB is a transcription factor consisting of homo- or heterodimers of Rel-proteins [127]. Within a dormant cell, the protein is localized in the cytoplasm and attached to the IκB inhibitor, thereby preventing its movement into the nucleus. Upon activation, IκB undergoes phosphorylation and is subsequently liberated from the complex. Following this, the Rel-proteins translocate to the nucleus and selectively bind consensus DNA elements to begin the expression of a series of genes associated with inflammation, hence leading to the creation of numerous mediators [128,129,130]. Many plant polysaccharides prevent the release of inflammatory mediators/or exert their anticancer activity by regulating the NF-κB pathway. An *Ophiopogonis Radix* polysaccharide extract was reported to inhibit the IKK-NF-κB pathway and islet inflammation induced by IL-1β [131]. In another study, the polysaccharide obtained from *Polygala tenuifolia* Willd. caused a reduction in the expression of NF-κB, thereby resulting in the reduced viability of OVCAR-3 cells [132]. Similarly, polysaccharides from *Hordeum vulgare L*. and *Portulaca oleracea* L. inhibited the nuclear translocation of NF-κB, thereby causing a reduction in human colon cancer cells (HT-29) and HeLa cells, respectively [77,133]. Figure 5 below summarizes the biological activities of plant polysaccharides.

## 5. Plant Polysaccharides’ Potential for Breast Cancer Treatment

In past decades, several plant polysaccharides, both in in vitro studies on various breast cancer cell lines and in vivo animal studies, have demonstrated their intriguing anticancer properties. A significant number of them have the ability to overcome the primary limitations of traditional chemotherapy. Plant polysaccharides derived from *Carthamus tinctorius L*, *Astragalus membranaceus*, *Morus alba L*, *Sparganii Rhizoma*, and *Solanum nigrum* have demonstrated specific cytotoxicity against breast cancer cells. These polysaccharides effectively inhibited the growth of breast cancer cells without causing the usual adverse effects associated with conventional treatment [134,135,136,137,138,139]. Luo et al. reported how *Carthamus tinctorius L*, safflower polysaccharide (SP) significantly and in a dose-dependent manner increased the rate of apoptosis in Luminal A subtype Human Epidermal Receptor-2 (HER-2) negative breast cancer cells (MCF-7) compared with untreated cells. The SP induced downregulation of B-cell lymphoma 2 (Bcl-2) expression while upregulating that of Bcl-2-associated X protein in MCF-7 cells. Furthermore, the treatment with the same SP caused a significant decrease in matrix metalloproteinase-9 expression; meanwhile, the tissue metalloproteinase-1 inhibitor expression was significantly upregulated in MCF-7 human breast cancer, indicating that SP exerted anti-metastatic activity on MCF-7 breast cancer cells [134]. Feng et al. in their study found the *Morus alba L.* polysaccharide to be significantly antioxidative against detrimental free radicals, and the polysaccharide significantly inhibited the proliferation, invasion, and migration of the Luminal A subtype-HER-2 negative MCF-7 breast cancer cells [138]. In the same vein, Yang et al. described how polysaccharide extracted from *Astragalus membranaceus* roots inhibited the migration and invasion in MCF-7 and MDA-MB-231 (triple-negative) breast cancer through the regulation of epithelial–mesenchymal transition (EMT) by the Wnt/β-catenin signalling pathway [137]. In the case of the *Solanum nigrum* polysaccharide (SnP), its tumour-suppressive ability was examined using the breast cancer animal model while monitoring the tumour weight and volume during the process. SnP was reported to significantly inhibit tumour weight and volume by 40% and 65%, respectively. This tumour-suppressive activity was realized to be through the immunomodulatory activity of the polysaccharide, as further investigation in the same experiment showed that SnP caused a significant increase in infiltrating natural killer (NK) cells, T cells, and macrophages in the treated tumour tissues. Furthermore, there was a significant increase in the concentrations of TNF-α, IFN-γ, and IL-4, while the concentration of IL-6 was reduced significantly in the serum of the treated breast cancer animal model. The results indicated that the tumour suppression mechanisms found in mice treated with SnP were likely to be a result of increasing the host immune response [135]. Polysaccharide fraction from Peony Seed Dreg was reported to cause cell cycle arrest in the G0/G1 phase. It caused significant downregulation of some cell cycle regulators (cyclin A, B1, D1, and E1; CDK-1, 2, 4, and 6; and p15, p16, p21, and p27); it also led to a significant change in expression of apoptotic and inflammatory promoting proteins (cytochrome C; Bax; Bcl-2; p-caspase-3, -8, and -9; and poly(ADP-ribose) polymerase (PARP)) in breast cancer MCF-7 [129]. Adewale et al. also reported antiproliferative and antioxidant screening of polysaccharides extracted from six different plants. In this study, a preliminary assessment was reported of the antiproliferative activities of various plant polysaccharide in breast cancer MCF-7 cells [102].

There is no doubt that plant polysaccharides do possess anticancer activity against breast cancer models both in vivo and in vitro. Common mechanisms of action through which these plant polysaccharides exhibit their antitumour activities include immunomodulation, anti-inflammatory activity, pro-apoptotic activity, and modulation of cell cycle progression by influencing the significant proteins in these pathways. Figure 6 below illustrates the effects of plant polysaccharides in breast cancer cells.

### 5.1. The Use of Plant Polysaccharides and Chemotherapy and Their Anticarcinogenic Mechanisms in Breast Cancer Studies

Given the adverse effects associated with chemotherapy, agents that can be used in combination with it to promote the anticancer efficacy while reducing or limiting its associated side effects, and thus improving the patient’s general condition, are usually of great benefits. The study of the combination of plant polysaccharides and chemotherapeutic drugs in breast cancer studies is a developing but interesting and promising field in cancer research. The enhancement of chemotherapy efficiency by plant polysaccharides could be at least in part due to their ability to facilitate the uptake of the drug.

Naturally, plant polysaccharides serve as storage and structural material and are relatively stable molecules. The stability, biodegradability, and lack of toxicity make plant polysaccharides excellent drug vehicles. Moreover, they can be chemically modified by sulfation, carboxymethylation, and phosphorylation, which can alter the plant polysaccharides’ biological properties, making them suitable as drug carriers [64]. Subsequently, these chemical alterations enhance the cellular uptake of the polysaccharide–drug conjugate [140,141] and its therapeutic efficiency in tissues [142]. Also, certain special polysaccharide properties, comprising enhancement in absorption, and chemical modification flexibility and biocompatibility, make them suitable for drug combination. These correspondingly enhance the target-specific delivery of the drug–polysaccharide conjugate to the required site, which in turn leads to a decreased concentration of the chemotherapeutic drug that is required for the expected treatment (compared to when the drug is not targeted to the right tissue). Subsequently, a decreased drug concentration targeted at the right tissue translates to a reduction in adverse side effects. Furthermore, evidence has shown that plant polysaccharides have modulatory effects on molecular targets that are also targets for chemotherapeutic drugs, thereby increasing the drug potency [116,143].

Based on the reported literature, the use of plant polysaccharides in tandem with chemotherapeutic drugs can take different forms:(i)Plant polysaccharides as vesicle for chemotherapeutic drug delivery [144,145].(ii)Plant polysaccharides used as conjugates to chemotherapeutic drugs [146].(iii)Plant polysaccharides administered in parallel with anticancer drugs [147].

#### 5.1.1. Plant Polysaccharides as Vehicle for Chemotherapeutic Drug Delivery 

Plant polysaccharides may be used to encapsulate or carry target chemotherapeutic drugs, thereby improving their delivery and efficacy. They can also be conjugated to drugs and then formulated into nanoparticles to deliver their contents. This approach takes into account the property of enhancement in adsorption of many polysaccharides, this allows the polysaccharide vesicles and their contents to pass through the membrane of the target cell without stress and elicit their action [148,149]. Raikwar and his colleagues reported specific plant polysaccharide-based nanoparticles in drug delivery and discussed the unique properties of these polymers, which make them suitable as drug carriers, including their significantly huge number of reactive groups, which allow for surface modification of the nanoparticles. Additionally, the mucoadhesive properties of polysaccharides can be harnessed to increase the retention time of drugs at the site of absorption [149]. Even though there are many studies on breast cancer cells exploring the use of chemotherapeutic drugs carriers based on polysaccharides derived from various sources, e.g., hyaluronic acid-chitosan nanoparticles, chitosan-dextran sulfate, wheat germ agglutinin-chitosan nanoparticles, and alginate [150,151,152,153], scarce reports on plant polysaccharide-based carriers for chemotherapeutic drugs delivery in breast cancer studies exist [144,145,146].

One of the studies on plant polysaccharide-based carriers for chemotherapy in breast cancer studies is that of Jang et al., who reported the evaluation made on the use of a *Pinus koraiensis* polysaccharide (PKP)-based injectable hydrogel as a drug carrier for paclitaxel (PTX) in breast cancer treatment. They showed that the PTX-loaded PKP hydrogel resulted in significant inhibition of 4T1 mouse breast cancer cells and MCF-7 human breast cancer cells growth, with respective inhibition rates of 71.26% and 77.38%. The PTX-loaded PKP hydrogel exhibited the lowest IC50 values (15.13 μg/mL for 4T1 and 10.32 μg/mL for MCF-7) when compared to free Taxol and PKP gel alone, indicating enhanced antiproliferative effectiveness. Interestingly, the PKP gel showed no toxicity to normal human liver cells (LO2), suggesting good biocompatibility. Furthermore, the in vivo studies confirmed the ability of the PTX-loaded PKP-based hydrogel to significantly inhibit tumour growth in the tumour-bearing mice compared to other treatment groups, including PTX and Taxol alone. Additionally, the study reported a synergistic effect of both PTX and PKP when used as PTX-loaded hydrogel. Specifically, the Combination Index (CI) values calculated by Chou–Talalay theory using the Compusyn or Calcusyn programs based on cell experiments for the PTX-loaded hydrogel indicated that both PTX and PKP act synergistically on both 4T1 and MCF-7 cancer cell lines. However, for the 4T1 cell line, synergism was observed at high doses, while the evidence of synergism in the MCF-7 cell line was across all doses for the PKP gel in combination with PTX [154]. The study generally highlights the prospects of PKP-based hydrogels as a promising plant polysaccharide-based drug carrier in the enhancement of chemotherapy efficacy in breast cancer while minimizing the systemic toxicity [144].

Similarly, in Xiong et al.’s study, *Rhodiola rosea* polysaccharides (RHPs) were used as a novel immunoactive vehicle for doxorubicin (DOX) delivery. RHP nanoparticles were created with folic acid (FA) and stearic acid (SA) for DOX delivery to a triple-negative breast cancer cell line, 4T1. This hydrophobic chemotherapeutic drug was enclosed within RHPs (FA-RHPs-SA), creating nanoparticles of approximately 196 nm in size and with pH-sensitive release properties to ensure the effectiveness in typical of tumour acidic environments.

In their experiments with cells, Xiong and colleagues demonstrated that 4T1 tumour cells uptook the DOX-FA-RHPs-SA nanoparticles, which also exhibited modulatory ability in tumour-associated macrophages (TAMs). The DOX-loaded nanoparticles specifically facilitated the M0 to M1 phenotypic transition in macrophages while enhancing the differentiation of M2 macrophages into M1, which are known for their antitumour activity. The antitumour effect of the DOXFA-RHPs-SA nanoparticles was further revealed in an animal study, where there was a higher antitumour effect of DOXFA-RHPs-SA nanoparticles compared to DOX monotherapy in the animal models. DOXFA-RHPs-SA nanoparticles reduced tumour size, induced tumour cell apoptosis, reduced collagen production, normalized vascular structure, and finally modulated immune function by enhancing the ratio of pro-inflammatory (Th1) cytokines to anti-inflammatory (Th2) cytokines [145]. The role of polysaccharides as promising candidates for drug delivery system is illustrated in Figure 7. 

#### 5.1.2. Plant Polysaccharides as Conjugates to Chemotherapeutic Drugs 

Plant polysaccharides can be chemically bonded to chemotherapeutic drugs, forming conjugates that improve the targeting of the drugs and reduce their toxicity. In their studies, Wang et al. focused on the formulation of a nano drug delivery system from a drug conjugate of AP-PP-DOX. AP-PP-DOX comprises the *Angelica sinensis* polysaccharide (AP), which is a traditional Chinese herbal medicine recognized for its immunomodulatory properties, and DOX conjugated through a peptide linker (PP) cleavable by matrixmetalloproteinase 2 (MMP2), which is often overexpressed in tumour tissues. This conjugate was then formulated into nanoparticles by self-assembly. DOX, which is hydrophobic, formed the core of the nanoparticle and hydrophilic AP formed the shell. In this system, AP acted both as a carrier for doxorubicin and an active therapeutic agent in breast cancer. The mechanism of this system is based on the fact that when AP-PP-DOX encounters MMP2 (originating from tumour cell), the peptide linker within the nanoparticles is cleaved, consequently leading to the controlled release of DOX and AP directly at the site of the tumour. The authors tested the potency of this AP-PP-DOX nano-system on a breast cancer cell line, MCF-7, with a considerable MMP2 expression and discovered that there was enhanced cytotoxicity in this cell line, which confirmed that the enzyme-sensitive design effectively triggers drug release in the tumour microenvironment. Additionally, immune responses were also modulated by the AP component of this system. Specifically, there was an increase in IL-2 and a decrease in IL-10 proteins, which are Th1-type and Th2-type cytokines, respectively, thus initiating a more favourable immune response against the tumour. Finally, the study concluded that the AP-PP-DOX nano delivery system offers a promising dual-action approach for cancer therapy, combining targeted chemotherapy with immune system enhancement to achieve an enhanced antitumour effect [146].

#### 5.1.3. Plant Polysaccharides Used in a Simultaneous Administration with Chemotherapeutic Drugs 

Plant polysaccharides can also be added simultaneously alongside chemotherapeutic drugs to enhance their therapeutic effects while reducing the toxic effects.

In the study reported by Yuan et al., selenium-containing polysaccharides (Se-PFPs) were extracted and purified from Se-enriched *Pyracantha fortunean.* The purified Se-PFPs comprise 93.7% carbohydrates, 2.1% uronic acid, and 3.7 μg/g of selenium and were classified as Se-conjugated polysaccharides.

These polysaccharides were administered alone or in combination with doxorubicin in vivo and in vitro to assess its antitumour-enhancing capacity. In vitro, Se-PFPs alone inhibited MDA-MB-231 cell growth via cell cycle arrest at the G2 phase through CDC25C-CyclinB1/CDC2 pathway inhibition. Se-PFPs in addition induced apoptosis via activation of the p53-mediated cytochrome c-caspase pathway. A combination of Se-PFPs and DOX resulted in enhanced antitumour effects on MDA-MB-231 cells. The combined treatment enhanced the cytotoxic effect of DOX alone, leading to significant cell growth inhibition and increased apoptosis. Se-PFPs in simultaneous combination with DOX further enhanced the activation of the p53-mediated cytochrome c-caspase pathway via the enhancement of pro-apoptotic proteins, comprising Puma, Noxa, and Bax coupled with a concomitant decrease in the anti-apoptotic protein Bcl2. The combination also led to enhanced cell cycle arrest at the G2 phase, when compared with individual components. 

In vivo, the combination of Se-PFPs with DOX further led to enhanced antitumour effects on MDA-MB-231-derived xenograft tumours in nude mice when compared with individual treatments. There was a significant reduction both in tumour volume and weight. Interestingly, the observed enhanced antitumour effects produced no noticeable adverse effects on the mice as they maintained stable body weights throughout the treatment duration. Aside from demonstrating potent antitumour activity, the combination of Se-PFPs with DOX also featured an ability to sensitize tumours to DOX, hence leading to enhanced tumour eradication. The absence of adverse effects shows the potential of Se-PFPs to enhance the therapeutic efficacy of DOX and maintain a safe status, thus offering a promising approach for improving breast cancer treatment outcomes [147].

A similar study by Sun et al. explored a novel approach in enhancing the efficacy of DOX in breast cancer study through combination with *Lycium barbarum* polysaccharides (LBP) and polypyrrole nanoparticles (PPY-NPs) under photothermal therapy. The LBP was administered simultaneously with DOX and PPY NPs to make the DOX + LBP + PPY NP combination. The objective for this study was to exploit the biological properties including the anti-inflammatory, antitumour, immune-boosting, and hepatoprotective properties of LBP, together with the photothermal capabilities of PPY NPs, in order to augment the antitumour potencies of DOX and reduce its systemic toxicity. The approach explores the complementary effects of herbal ingredients and nanotechnology to achieve enhanced tumour suppression and reduced toxicity. 

In vitro, the DOX + LBP + PPY NP combination, in the presence of near-infrared (NIR) laser irradiation, inhibited 4T1 breast cancer cell growth by eight times more, compared to DOX alone. In vivo, the findings revealed that the combination therapy achieved an 87.86% tumour inhibition rate. The addition of LBP and PPY NPs improved DOX accumulation at the tumour site, enhanced its antitumour effects, and also reduced its toxic side effects in terms of liver damage and systemic inflammation. Finally, the simultaneous administration of DOX, LBP, and PPY NPs, together with photothermal therapy, represents a promising strategy for breast cancer treatment. The results indicate that LBP enhances the antitumour effects of DOX by modulating the immune response, decreasing immunosuppressive factors including IL-10 and IgA, while increasing pro-inflammatory cytokines (IFN-γ and TNF-α). This potentiation of the immune response, combined with the localized drug delivery and photothermal effects provided by PPY NPs, makes this combination therapy a potential candidate for further clinical investigation in cancer treatment [155]. 

Galactomannan polysaccharide from guar gum (*Cyamopsis tetragonoloba*) was also employed in simultaneous administration with 5-fluorouracil, doxorubicin, irinotecan, and cisplatin separately to immune-compromised mice bearing mammary tumour ZR-75-1. There was no cytotoxicity in the mice, as evidenced by stable body weight growth, but a noticeable efficacy enhancement was observed in the antitumour effects of the tested chemotherapeutic drugs [156]. The modified polysaccharide used here was DAVANAT, which was obtained from the natural (‘initial’), very -large-molecular-weight, and poorly water-soluble galactomannan. DAVANAT was obtained through a controlled molecular-weight reduction, which enhanced its water solubility leading to better anticancer effect [156]. 

The impact of the simultaneous administration of polysaccharides from sweet green pepper (*Capsicum annuum*, CAP) and methotrexate (MTX) on mammary tumours in vivo and in vitro was similarly investigated. Alone, treatment with CAP (100 mg/kg, p.o.) reduced tumour growth for over 31 days without induction of toxicity. When combined with MTX (1 mg/kg, i.p.) for 21 days, CAP enhanced tumour growth inhibition to 95%, compared to the efficacy of MTX alone. Either alone or with MTX, CAP treatment decreased pro-inflammatory cytokines (IL-4 and IL-10) and increased IL-6 and TNF-α levels, indicating a shift towards an inflammatory response while reducing the expression of angiogenesis markers (VEGF) in tumours. CAP also showed enhanced cytotoxic effects with MTX against mammary tumour cell lines (MDA-MB-231 and MDA-MB-436) and Ehrlich tumours. These findings suggest CAP’s potential as a low-toxicity adjunct to chemotherapy, with its effects attributed to the modulation of inflammation and angiogenesis [157]. 

Lastly, the study of Shang et al. explored the simultaneous administration of *Tetrastigma hemsleyanum* polysaccharides (THPs) and DOX in triple-negative breast cancer (TNBC). This combination enhanced the efficacy of DOX against TNBC, meanwhile reducing its toxicity to the heart, liver, and kidneys. The mode of action of THP is by induction of ferroptosis and ferritinophagy via the xCT/GSH/GPX4 and Nrf2/NCOA4/FTH1 pathways, with its branched-chain hexose being involved in direct binding to the xCT protein to inhibit its expression. Furthermore, THP increases the ratio of CD4^+^ and CD8^+^ T cells to regulatory T cells [158]. The effects of plant polysaccharides and chemotherapy in breast cancer studies are gathered in Table 1. 

In the studies described above, the significant potential of plant polysaccharides in the improvement of the therapeutic efficacy and in reducing the toxicity of chemotherapeutic drugs in breast cancer treatment is provided. The information provided was based on combined effects of plants polysaccharides with paclitaxel, doxorubicin, 5-fluorouracil, irinotecan, cisplatin, and methotrexate (representing all the major classes of chemotherapeutic agents). Particularly, doxorubicin was commonly used in these studies, with only one study each reporting the effects of PTX, fluorouracil, irinotecan, cisplatin, and methotrexate with plant polysaccharides. To the best of our literature search and at the time of this report, only polysaccharides from *Angelica sinensis* [87], *Lycium barbarum* [91], *Pinus koraiensis* [144], *Rhodiola rosea* [145], *Pyracantha fortunean* [147], *Cyamopsis tetragonoloba* [156], *Capsicum annuum* [157], and *Tetrastigma hemsleyanum* [158] were reported for their combined effects with chemotherapeutic drugs in breast cancer preclinical studies. 

The molecular size, or more specifically the molecular weight of polysaccharides, is one of the major determinants of their biological activities [159] including antitumour activities, although the structure–activity relationship is unclear. Polysaccharides extracted from natural sources can have broad or mixed molecular weights (mixture of high- and low-molecular-weight molecules), which can influence their anticancer capacities. For example, some higher-molecular-weight glucans have been reported to have higher antitumour activities over the lower-molecular-weight ones; this quality has been attributed to the fact that high-molecular-weight polysaccharides have a better binding affinity toward the receptors of the immune cells and therefore can activate the innate and adaptive immune response, which is important in tumour elimination [160]. On the other hand, this immunostimulatory effect can lead to adverse effects, such as inflammation or autoimmune reactions, which may compromise patient tolerance to the therapy. Large polysaccharides recognized by the immune system as foreign may be cleared from circulation, which could reduce their effectiveness either alone or in combination with anticancer drug. Meanwhile, another study confirmed that lower-molecular-weight fucoidan (obtained through enzymatic degradation of higher-molecular-weight polysaccharides) exhibited greater antitumour potential compared to higher-molecular-weight polysaccharides [161]. Low-molecular-weight polysaccharides have been confirmed to have the ability to penetrate and stimulate the concerned cells to exhibit their antitumour potential [162]. 

Overall, molecular weight or size of the polysaccharides can affect their antitumour potential whether alone or in combination with existing anticancer drugs, as mentioned earlier for DAVANAT in combination with anticancer drugs [156]. Future research directions have been suggested for further investigation into the relationship between polysaccharide molecular weight and antitumour activity to gain a better understanding of this context.

There is an obvious gap in the study of more plant polysaccharides for their combined anticarcinogenic effects with conventional chemotherapeutic drugs. This review highlights that plant polysaccharides have the potential ability to enhance cytotoxic activities of major categories of chemotherapeutic drugs while suppressing their associated toxicities in pre-clinical studies, some of which are already undergoing clinical trials. Therefore, further exploration of more plant polysaccharides is warranted to fully explain their potential benefits in combination with chemotherapy for breast cancer treatment. By expanding the scope of research to include more plant-derived polysaccharides, it fosters our understanding on how to harness their therapeutic potential and improve treatment outcomes for breast cancer patients. 

## 6. Future Directions

There are undoubtedly promising findings from studies on the enhancement of breast cancer chemotherapy efficacies by polysaccharides; however, the research on the combination of plant polysaccharides and chemotherapy in breast cancer remains limited. The limitations in the use of plant polysaccharides in clinical studies are numerous, including lack of clarification on their bioavailability, variability in effectiveness, lack of standardization in plant sources, unknown molecular mechanisms of cytotoxicity, the molecular targets, etc. Moreover, the current research methods on their bioactive and antitumour properties are insufficient.

Just a few studies regarding the use of plant polysaccharides and chemotherapy, which were reported in this review, are very promising, and they are pointers that plant polysaccharides can indeed enhance the efficacies of chemotherapies while suppressing the associated toxicities. In a way, to improve the use of plant polysaccharides in combination with chemotherapeutic drugs, optimization of extraction methods may be necessary for better consistency and standardization of the plant polysaccharide drugs [163]. Additionally, investigating the synergy between plant polysaccharides from various sources and chemotherapeutic agents is crucial for advancing this therapeutic approach in breast cancer treatment. In this review, only one study reported on the synergistic interaction between plant polysaccharides and chemotherapy; therefore, there is a need for more studies on the investigation of synergistic interaction between plant polysaccharides and chemotherapy since this information could be valuable for the development of therapeutic combinations in breast cancer treatment. Future studies should also focus on detailed mechanistic investigations to uncover more molecular pathways via which plant polysaccharides enhance chemotherapy efficacy and mitigate its side effects. This information will go a long way in contributing to the body of scientific knowledge and providing a detailed scientific information to assist in developing effective plant polysaccharide-chemotherapy formulation. The understanding of these mechanisms can also aid in harnessing the potential of plant polysaccharides, leading to more potent and safer breast cancer treatments. The exploration of the bioactive properties of other natural compounds as well could be of great benefit in anti-cancer therapies.

## 7. Conclusions

This review highlights the importance of plant polysaccharides in the enhancement of chemotherapy effectiveness and reduction in its toxic side effects in breast cancer. The combination of plant polysaccharides and chemotherapy in breast cancer treatment holds great significance. However, further research is needed to explore the synergy between more plant polysaccharides and chemotherapeutic agents and to understand the underlying molecular mechanisms behind this synergy.

## Figures and Tables

**Figure 1 ijms-25-12202-f001:**
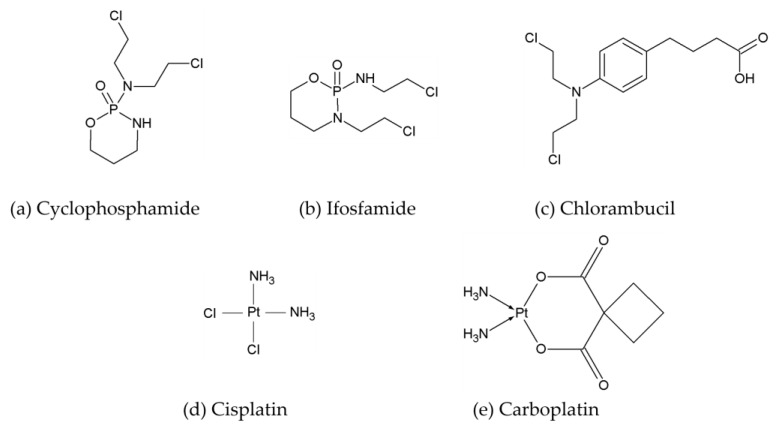
Exemplary structures of alkylating agents used in breast cancer treatment.

**Figure 2 ijms-25-12202-f002:**
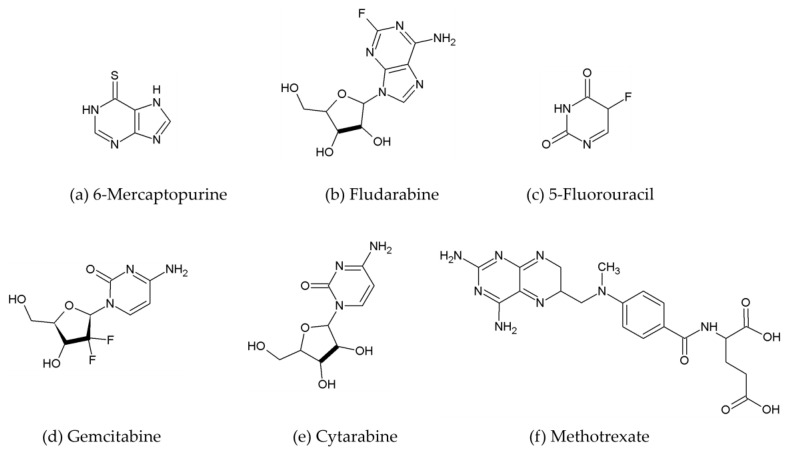
Exemplary structures of antimetabolites used in breast cancer treatment.

**Figure 3 ijms-25-12202-f003:**
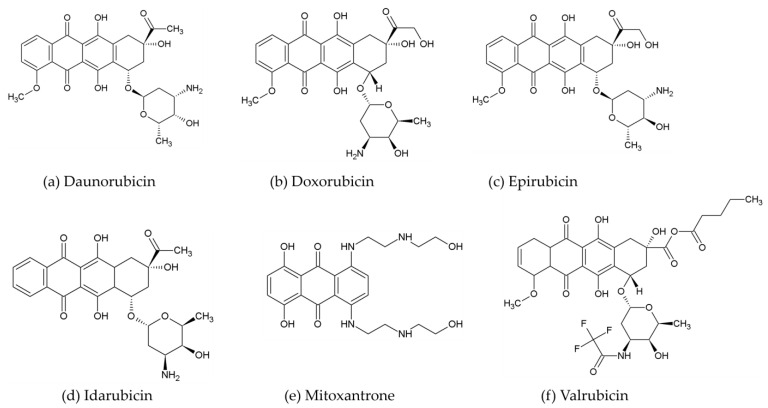
Exemplary structures of anthracyclines used in breast cancer treatment.

**Figure 4 ijms-25-12202-f004:**
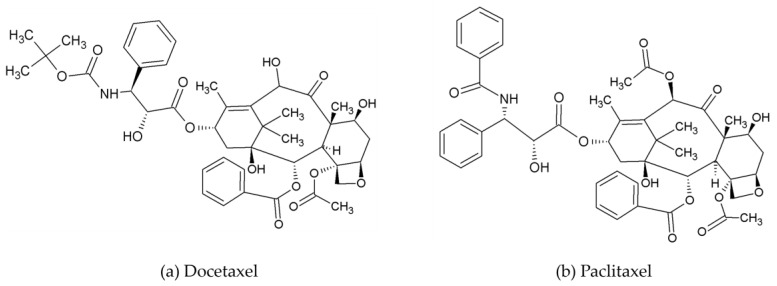
Exemplary structures of microtubule inhibitors used in breast cancer treatment.

**Figure 5 ijms-25-12202-f005:**
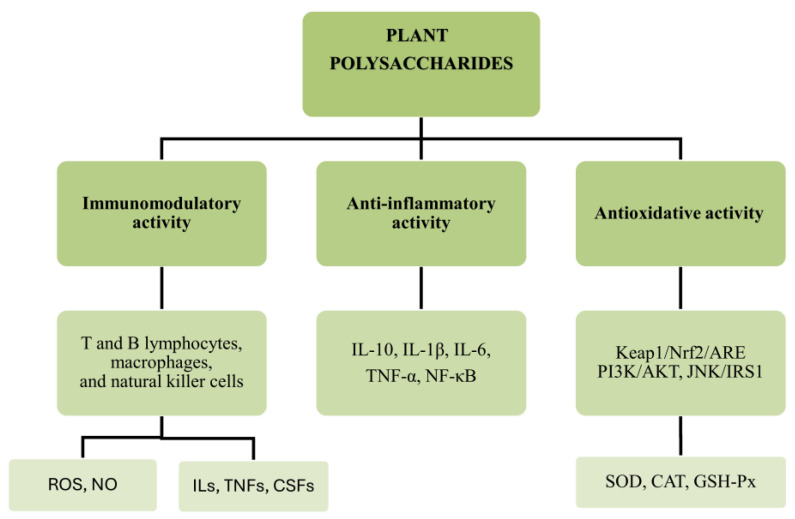
Biological activities of plant polysaccharides. This highlights the roles of plant polysaccharides in three main areas: immunomodulatory, anti-inflammatory, and antioxidative activities. Plant polysaccharides stimulate immune responses through T and B lymphocytes, macrophages, and natural killer cells, which lead in turn to the regulation of the immune mediators like ROS and NO, and also the regulation of ILs, TNFs, and CSFs. They mediate anti-inflammatory responses via cytokines and through the NF-κB pathway. Their influence on antioxidative activity is by activation of cellular pathways like Keap1/Nrf2/ARE, PI3K/AKT, and JNK/IRS1, which lead to the enhanced activity of antioxidant enzymes. CAT, catalase; CSF, colony-stimulating factor; GSH-Px, glutathione peroxidase; IL, interleukin; NF-κB, nuclear transcription factor κB; NO, nitric oxide; ROS, reactive oxygen species; SOD, superoxide dismutase; TNF, tumour necrosis factor.

**Figure 6 ijms-25-12202-f006:**
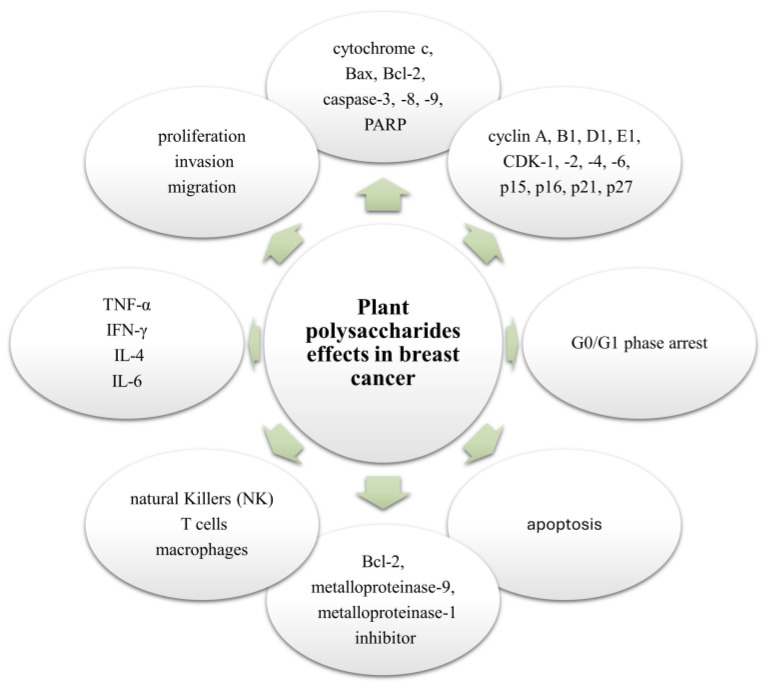
The effect of polysaccharides on breast cancer cells. This figure emphasizes the various biological processes that plant polysaccharides target in breast cancer. These main biological processes include proliferation, invasion, and migration by regulating associated protein like cytochrome C; Bax; Bcl-2; caspase-3, -8, and -9; and PARP. They also stimulate G0/G1 phase arrest through the modulation of cyclins (A, B1, D1, and E1) and CDKs (1, 2, 4, and 6), along with p15, p16, p21, and p27. Furthermore, they promote apoptosis through Bcl-2 and metalloproteinases inhibitor expression and improve immune activities through the regulation of natural killer cells, T cells, macrophages, and cytokines (TNF-α, IFN-γ, IL-4, and IL-6). Bax and Bcl2, apoptosis-modulating proteins; CDK, cyclin-dependent kinase; IFN-ϒ, interferon ϒ; IL, interleukin; p15, p16, p21, and p27, cyclin-dependent kinase inhibitors; PARP, poly(ADP-ribose) polymerase; TNF-α, tumour necrosis factor α.

**Figure 7 ijms-25-12202-f007:**
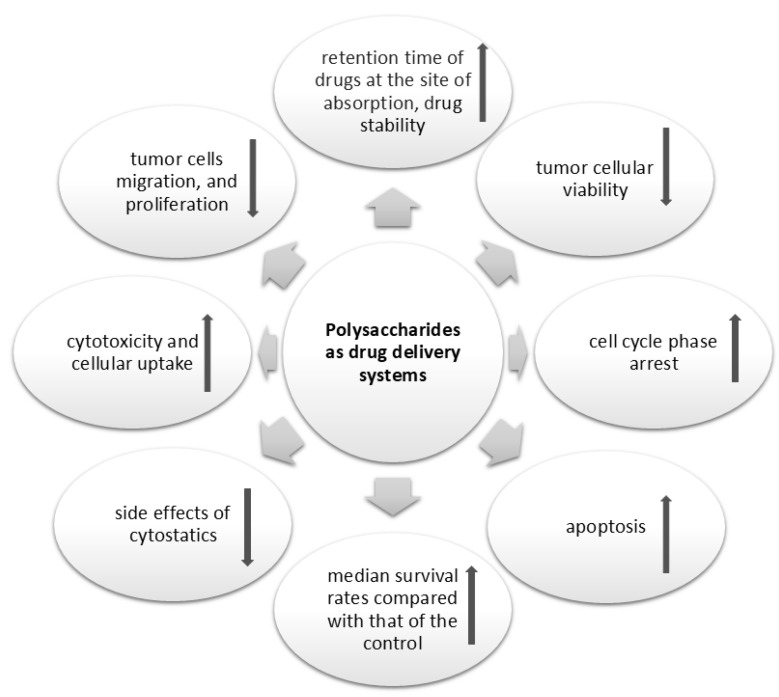
An illustration of polysaccharides as promising candidates for drug delivery system due to the ability to encapsulate chemotherapeutic agents with subsequent formation of nanoparticles and hydrogels. This facilitates drug stability, synergistic interaction, and targeted drug delivery, all of which enhance therapeutic efficacy and reduce side effects.

**Table 1 ijms-25-12202-t001:** The effects of plant polysaccharides and chemotherapy in breast cancer cell studies.

Polysaccharide	Chemotherapy	Experimental Model	Outcomes	References
**1. Plant Polysaccharides as vesicles for chemotherapeutic drug delivery**				
*Pinus koraiensis* polysaccharide (PKP)	Paclitaxel	In vivo: 4T1 mouse breast cancer cells In vitro: 4T1 and MCF-7 human breast cancer cells	- Enhanced antiproliferative effectiveness - Synergistic effect with PTX - Significant tumour growth inhibition - No toxicity to normal human liver cells	[144]
*Rhodiola rosea* polysaccharides (RHPs)	Doxorubicin	In vivo: 4T1 tumour-bearing mice In vitro: 4T1 tumour cells	- pH-sensitive release in acidic tumour environments - Modulation of tumour-associated macrophages (TAMs) - Enhanced M0 to M1 transition in macrophages - Reduced tumour size and induced apoptosis	[145]
**2. Plant Polysaccharides in Conjugation with Chemotherapeutic Drugs**				
*Angelica sinensis* polysaccharide (AP)	Doxorubicin	MCF-7 breast cancer cells	- Targeted drug release via MMP2 cleavage - Enhanced cytotoxicity - Modulated immune responses (increased IL-2, decreased IL-10)	[146]
**3. Plant Polysaccharides in Simultaneous Administration with Chemotherapeutic Drugs**				
Selenium-containing polysaccharides (Se-PFPs) from *Pyracantha fortunean*	Doxorubicin	In vivo: MDA-MB-231-derived xenograft tumours in nude mice In vitro: MDA-MB-231 cells	- Induced apoptosis via p53-mediated pathway - Enhanced cell cycle arrest at G2 phase - Increased pro-apoptotic proteins - Sensitized tumours to DOX	[147]
*Lycium barbarum* polysaccharides (LBPs)	Doxorubicin	In vivo: Mice bearing 4T1 cells In vitro: 4T1 breast cancer cells	- Enhanced antitumour effects under photothermal therapy - Improved drug accumulation - Reduced systemic toxicity - Modulated immune response	[155]
Galactomannan polysaccharide from *Cyamopsis tetragonoloba*	5-Fluorouracil, doxorubicin, irinotecan, cisplatin	Nude mice injected with ZR-75-1 mammary tumour cells	- Enhanced efficacy in breast cancer models - No cytotoxicity observed	[156]
*Capsicum annuum* polysaccharides (CAP)	Methotrexate	In vivo: Mice bearing Erlich tumour, In vitro: MDA-MB-231, MDA-MB-436	- Significant tumour growth inhibition	[157]
*Tetrastigma hemsleyanum* polysaccharides (THPs)	Doxorubicin	Triple-negative breast cancer (TNBC) model	- Induction of ferroptosis and ferritinophagy - Increased CD4^+^ and CD8^+^ T cells - Reduced toxicity to heart, liver, and kidneys	[158]

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
