# Peer review of "The Potential of Plant Polysaccharides and Chemotherapeutic Drug Combinations in the Suppression of Breast Cancer"

_ijms, 2024, doi:10.3390/ijms252212202_

Round 1
Reviewer 1 Report
Comments and Suggestions for Authors
The authors discuss the challenges of chemotherapy, noting its dangerous side effects due to non-specific targeting, poor therapeutic indices, and high dosage requirements. They explore the potential of natural compounds, particularly plant polysaccharides, which are biologically active and safe, to enhance chemotherapy's cytotoxicity against cancer cells while reducing its side effects. The review highlights how combining plant polysaccharides with conventional chemotherapy can offer synergistic benefits, lowering drug dosages and improving therapeutic outcomes, especially in breast cancer treatment. The findings emphasize the potential of these natural compounds to make cancer therapies more effective and safer.
Here are some suggestions for improving the text and making the review more comprehensive:
1. Clarify Key Points
- Problem Statement: The introduction mentions breast cancer statistics but could better emphasize the need for new therapeutic strategies early on, linking directly to the benefits of plant polysaccharides.
- Natural Compounds: When discussing natural compounds like polysaccharides, clearly state what makes them unique or superior compared to other adjunct therapies (e.g., molecular structures or specific mechanisms that help in chemotherapy).
2. Improve Structure
- Subheadings: Adding clear subheadings throughout the text will improve readability. For instance, separate sections for "Chemotherapy Challenges," "Benefits of Plant Polysaccharides," and "Synergistic Effects with Chemotherapy."
- Flow: The transition from chemotherapy's limitations to the potential of polysaccharides could be smoother. Use phrases like “In light of these challenges, the focus has shifted to natural agents…” to make the shift clearer.
3. Data and Examples
- Statistics: Provide updated, specific data where possible. The review could strengthen its arguments with more precise statistics, for example, detailing how polysaccharides have improved survival rates or reduced side effects in clinical or preclinical studies.
- Case Studies: Add examples of successful trials or studies involving plant polysaccharides combined with chemotherapy. This would solidify the claim of their therapeutic potential.
4. Expand on Mechanisms
- The mechanisms of how polysaccharides synergize with chemotherapy (e.g., enhancing drug sensitivity, modulating immune responses) could be expanded with more specific examples or molecular pathways.
5. Address Limitations
- The review should discuss the limitations of using plant polysaccharides (e.g., variability in effectiveness, lack of standardization in plant sources) and mention ongoing challenges in clinical translation.
6. Suggestions for Future Research
- Propose future research directions, such as exploring other natural compounds or optimizing polysaccharide extraction methods for better consistency.
General Suggestions for Improved Review:
- Depth and Breadth: While the current review touches on key aspects, it could benefit from exploring additional plant-based compounds beyond polysaccharides (e.g., alkaloids or terpenoids).
- Visual Aids: Consider including diagrams or charts summarizing the mechanisms of chemotherapy and polysaccharide interactions to help readers visualize complex processes.
- Literature Review: Ensure that the review includes recent studies (2020-2024) on natural compounds in breast cancer treatment to maintain relevance.
Incorporating these suggestions will make the review more thorough, focused, and impactful, increasing its contribution to the field.
Finally, advancing breast cancer treatments. PD-L1 targeting and microRNA therapies offer new hope for aggressive cancers like triple-negative breast cancer (TNBC), which lack effective traditional therapies. Biomimetic carriers such as leukosomes aim to improve drug delivery and reduce side effects, addressing the review's call for less toxic therapies. Personalized medicine and single-cell sequencing enable more precise, targeted approaches, crucial for overcoming breast cancer's complexity and drug resistance (please refer to PMID: 34440380 and expand the introduction and the discussion sections).
Comments on the Quality of English LanguageThe authors address the challenges of chemotherapy, particularly its non-specific targeting, high dosage requirements, and toxic side effects. They highlight the potential of natural compounds, especially plant polysaccharides, which are biologically active and safer, to enhance chemotherapy’s effectiveness against cancer cells while reducing toxicity. Combining plant polysaccharides with chemotherapy is discussed as a strategy to lower drug doses and improve therapeutic outcomes in breast cancer treatment.
To enhance the review, it is recommended to:
- Clarify key points, such as emphasizing the need for new therapies and detailing the unique benefits of natural compounds.
- Improve structure with subheadings and smoother transitions.
- Include data, case studies, and more examples of successful treatments involving polysaccharides.
- Expand on the mechanisms of action and address the limitations of natural compounds.
- Suggest future research directions, such as exploring other natural compounds.
Reviewer 2 Report
Comments and Suggestions for Authors
The review article on plant polysaccharides and their possible use in breast cancer treatment is interesting. The rationale is clear. However, there are some points to add to the manuscript. The first issue is that the text in sections 2.3.2, 2.3.4, and 2.4 is too short for the review's aim. I also think that Taxol and inhibitors of estrogen receptors are not described. How do the authors acknowledge the possible role of phytoestrogens in the review? I do not see in the review stating how specific glycan structures can activate immune cells ie, alpha galactosylceramide and how glycan structures can block immune cells' activity, ie, https://doi.org/10.1016/j.celrep.2024.114105 https://doi.org/10.1016/j.trecan.202 1.08.001. Finally, do the authors envision polysaccharides as good delivery systems that must be rewritten; a Figure will be appropriate.
Round 2
Reviewer 2 Report
Comments and Suggestions for Authors
The authors made minor cosmetic changes in the manuscript and did not respond to the queries raised. There is no point in discussing known mechanisms of drugs that are already on the market. There is no reasonable comparison between structures, modified structures, and new delivery systems of some compounds in which some activity was defined in vitro.
Author Response
Comment 1: The authors made minor cosmetic changes in the manuscript and did not respond to the queries raised.
Response1: Thank you for your feedback about our response and apologize if you we did not meet your expectations. We aim to provide a more comprehensive response to satisfactorily address your queries in this revision round.
Comment 1: The authors made minor cosmetic changes in the manuscript and did not respond to the queries raised.
Response 1: Thank you for your feedback about our response and apologize if you we did not meet your expectations. We aim to provide a more comprehensive response to satisfactorily address your queries in our next revision.
Comment 2: There is no point in discussing known mechanisms of drugs that are already on the market.
Response 2: We appreciate you for the thoughtful feedback. Thank you for your perspective regarding the inclusion of the mechanisms of the chemotherapeutic drugs which are already on the market. The reason for including these mechanisms is to reinforce what is known and to show the similarity between the mechanism of these conventional drugs and the plant polysaccharides that are our focus. We intended to avoid an elaborate discussion on this subject since it is not our focus, but we believe a concise overview is critical to provide readers with relevant context. Interestingly, another reviewer previously supported this inclusion to strengthen our argument, he even suggested we provide full details but we explained that we provided limited information because it was not our major focus. Therefore, we choose to maintain a balanced level of detail. We appreciate you once again for your insights.
Comment 3: There is no reasonable comparison between structures, modified structures, and new delivery systems of some compounds in which some activity was defined in vitro.
Response 3: We thank the reviewer for this insightful feedback. We as well acknowledge the germane nature of the comparison of structures and delivery systems to emphasize compound activity. Nevertheless, main focus of our manuscript as mentioned also in our abstract is about the therapeutic potential of combination of chemotherapy with plant polysaccharides in breast cancer. We aimed to discuss the implications of this combination for improvement of chemotherapeutic efficacy, instead of providing a detailed comparative analysis of various structures or delivery systems. In recent years there are many review articles available describing the structures of polysaccharides (for example: DOI: 10.1201/9780203485286.ch3, doi:10.3390/molecules24173122, DOI 10.3389/fnut.2022.1021448, DOI: 10.3389/fphar.2021.767947), including plant polysaccharides, so we cite only selected publications. Even though such comparisons could be informative, they do not fall within the scope of the objectives of our review.
We hope we have effectively communicated our position and addressed your concerns. Thank you for your consideration.
Round 3
Reviewer 2 Report
Comments and Suggestions for Authors
Thank you for adding more details to some of the issues raised, although I think the last part of your response concerning drug delivery should have been discussed further, as the authors are well aware. I will not delay the manuscript longer.
